# P*I*PA: AN AGENT FOR PROTEIN INTERACTION IDENTIFICATION AND PERTURBATION ANALYSIS

## ABSTRACT

Protein–protein interactions (PPIs) play a fundamental role in the functioning of proteins and the formation of cellular pathways. Given their implication in numerous disease processes, PPIs represent important targets for therapeutic intervention. To enhance the efficacy and efficiency of PPI identification, pathway analysis, and disease target screening, we present PIPA, a tool-augmented intelligent agent that automates the entire PPI-centric discovery pipeline through a structured **Plan-Select-Execute-Reflect** loop, to assist biological researchers in discovering novel PPIs and predicting the effects caused by pathological mutations. PIPA integrates multiple tasks, including automated database retrieval, protein interaction identification, pathological mutation annotation, and interface perturbation prediction. To test its scientific discovery capabilities, PIPA was used to investigate the interaction between the proteins of the endoplasmic reticulum (ER) and Dynamin 2 (DNM2). The identified hypo-PPI with dynamin mutation, a potential target for therapeutic intervention of centronuclear myopathy, was experimentally validated in the wet lab. Furthermore, PIPA autonomously explored numerous PPI and mutation combinations to elucidate disease-related alterations in protein–protein interactions at the proteome scale. The scientific data used in this study has been developed into a benchmark named ER-MITO. As a novel framework for quantitatively evaluating agent capabilities on biologically meaningful tasks, ER-MITO benchmark moves beyond simplistic QA metrics to assess an agent's skill in (i) predicting novel inter-organelle PPIs and (ii) accurately retrieving and associating pathogenic mutations with proteins, further facilitating the evaluation of other AI systems and agents in terms of their capabilities for scientific discovery.

keywords: Scientific discovery agent, Cellular organelle, Protein-protein interaction, Large language model

## 1 INTRODUCTION

Understanding their structural basis and the effects of pathogenic mutations on interaction interfaces is therefore critical for elucidating disease mechanisms and developing targeted therapeutics. The field was revolutionized by AlphaFold2 (AF2), which achieved near-experimental accuracy in monomeric protein structure prediction Jumper et al. (2021a). This breakthrough was rapidly extended to complexes via AlphaFold-Multimer Evans et al. (2022) and related approaches like AF2Complex Gao et al. (2022a), shifting the community's focus from obtaining structures to effectively utilizing them for large-scale, systematic PPI analysis—including interaction validation, interface characterization, and mutational impact assessment.

While AF2 and its derivatives provide high-fidelity structural hypotheses, leveraging them for proteome-wide or mutation-aware studies presents a new set of challenges. Current computational strategies often rely on fragmented, manual workflows that combine standalone tools. These include methods for complex modeling Gao et al. (2022a), interaction confidence estimation (e.g., pDockQ Bryant et al. (2022a)), and functional annotation Bell et al. (2022b). Subsequent feature-based machine learning approaches, such as PPIscreenML Mischley et al. (2024) and the Predictomes framework Schmid and Walter (2025), have demonstrated the utility of AF2-derived structural features for distinguishing biologically relevant interactions from false positives on a large scale. Meanwhile, deep learning architectures have advanced residue-level interface prediction using graph neural net-

works Mahbub and Bayzid (2022); Réau et al. (2023) and multimodal approaches that fuse language models with structural insights Jha et al. (2023); Liu et al. (2025).

Despite these advancements, a significant gap remains in creating integrated and autonomous systems that can seamlessly traverse the entire discovery pipeline—from candidate prioritization and structural prediction to mutation mapping and pathological perturbation analysis. This fragmentation hinders reproducibility, scalability, and the dynamic incorporation of new evidence, particularly from the vast and growing literature on genetic variants Drake et al. (2025). The emerging paradigm of agentic scientific discovery Correia et al. (2025); Lee et al. (2025), powered by improved reasoning capabilities in large language models OpenAI (2025), offers a promising path forward. These systems aim to orchestrate specialized tools and knowledge sources within a cohesive, goal-directed framework.

To address this gap, we introduce PIPA (Protein Interaction Identification and Perturbation Analysis), a tool-augmented agent that automates the end-to-end analysis of PPIs and their pathological perturbations, integrating AF2-based structural modeling with automated database mining, literature-scale evidence retrieval, and biophysical interface analysis 2. Its key contributions include: (1) an agent architecture that dynamically combines general-purpose reasoning models (Intent Recognition, Reasoning & Execution, Reflection, Summarization) with domain-specific tools in a unified planning–select-execution–reflection loop; (2) hierarchical tool abstraction and orchestration that allow the agent to dynamically select the most appropriate tool based on the data modality required and generate executable steps that are context-aware of the specific biological tools and databases available; and (3) pioneering automated benchmarking for rigorous AI for science evaluation by a standardized, reproducible testbed for comparing different AI agents on their scientific reasoning ability, also a challenge that general-purpose agents like BiomniLee et al. (2025) have not yet fully tackled.

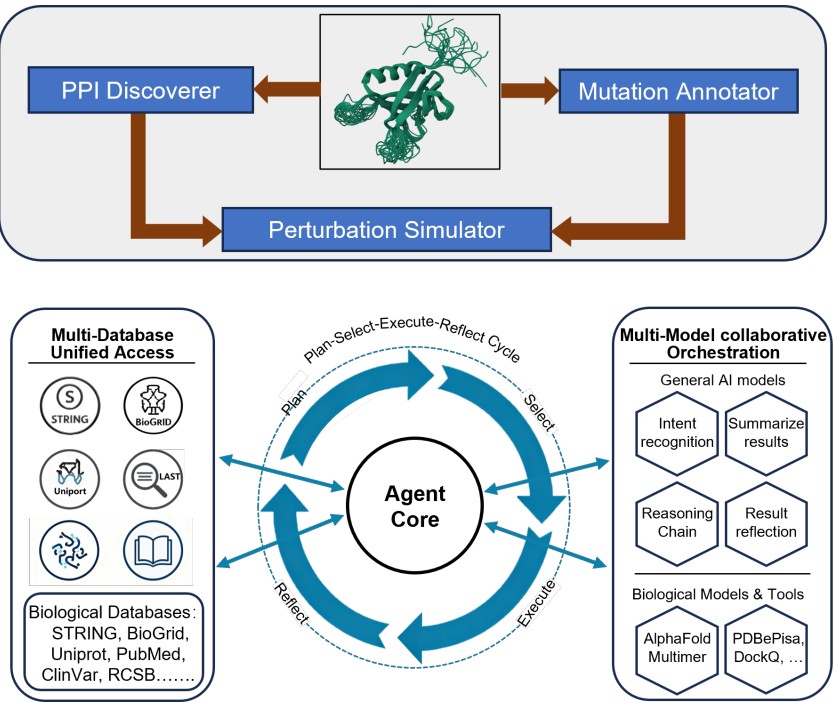

Figure 1: Agents architecture

## 2 METHODS

### 2.1 OVERVIEW OF PIPA

We introduce PIPA, an intelligent agent framework that automates the systematic discovery and characterization of protein-protein interactions (PPIs) and their pathological perturbations. PIPA integrates structural bioinformatics, database mining, and mutational analysis through a unified tool-augmented architecture. The system orchestrates multiple specialized components including AlphaFold-Multimer for complex structure prediction, PDBePISA for interface characterization, BLASTp for homology-based candidate expansion, and comprehensive biological databases (BioGrid, StringDB, UniProt, etc.) for evidence integration.

PIPA operates through three sequential analytical stages: (1) *PPI Mining and Reliability Assessment* identifies interaction candidates from experimental databases and homology inference, then validates them using structural metrics pDockQ; (2) *Pathological Mutation Integration* aggregates disease-associated variants from literature and clinical databases into a structured protein–mutation–disease association matrix; (3) *Interface Perturbation Analysis* quantifies mutational effects on binding interfaces through comparative biophysical profiling of wild-type versus mutant complexes. This pipeline enables automated prioritization of PPIs with potential disease relevance based on integrated confidence scores and perturbation severity metrics.

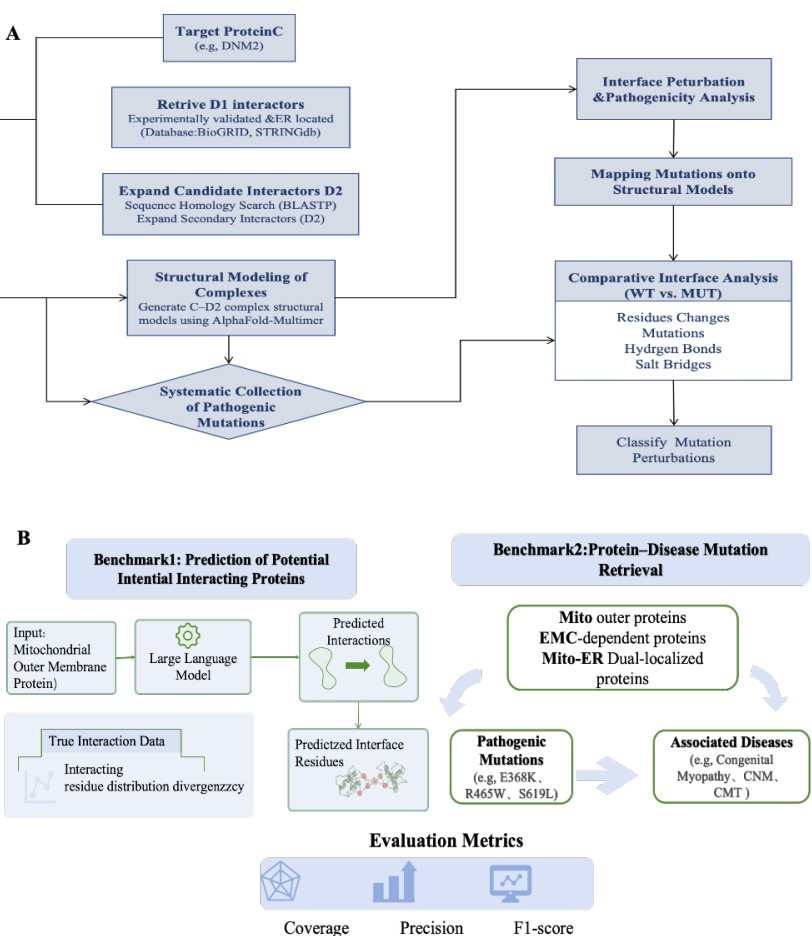

Figure 2: Overall workflow of the PIPA system for identifying disease related PPI

## 2.2 AGENT ARCHITECTURE

PIPA's architecture implements a tool-augmented agent paradigm specifically designed for biomedical discovery tasks. The framework consists of four modular layers: (1) a *general-purpose reasoning engine* comprising four LLM-based components for intent recognition, chain-of-thought reasoning, result reflection, and summary generation; (2) a *specialized execution layer* integrating domain-specific tools including AlphaFold-Multimer for high-accuracy complex structure prediction; (3) a *knowledge retrieval layer* providing access to millions of biomedical publications through semantic search; and (4) a *unified tool interface* abstracting heterogeneous biological databases and computational tools into standardized API calls.

### 2.2.1 EXECUTION LOOP: PLAN-SELECT-EXECUTE-REFLECT

PIPA operates through an iterative Plan-Select-Execute-Reflect cycle that ensures robust task completion. The *Plan* phase involves the reasoning engine decomposing user queries into executable workflows, determining required tools and data sources. During *Select*, the agent dynamically chooses appropriate tools based on task characteristics (e.g., interface analysis vs. structural prediction). The *Execute* phase invokes selected tools and aggregates their outputs. Finally, *Reflect* evaluates result coherence and task satisfaction, triggering iterative refinement when necessary.

This cyclic execution model provides several advantages: (1) transparency through explicit planning and selection rationales; (2) adaptability to heterogeneous biological questions; (3) fault tolerance through reflective validation. The framework supports seamless integration of additional tools and data sources, making it extensible to emerging analytical methods. By combining structured biological knowledge with flexible reasoning capabilities, PIPA establishes a new paradigm for automated protein interaction discovery and characterization.

### 2.2.2 TOOL ORCHESTRATION

A significant challenge in scientific AI is the heterogeneity of data sources and computational tools. PIPA introduces a hierarchical tool abstraction layer that goes beyond simple API calls. PIPA abstracts biological databases and tools into composable modules with standardized input-output specifications (1). Database interfaces include STRING (protein interaction networks), BioGRID (experimentally validated interactions), UniProt (sequence annotations), PDB (structural data), and PubMed (literature access). Computational tools encompass BLAST (sequence similarity), PDBePISA (interface analysis), and DockQ (complex quality assessment). This abstraction enables dynamic tool selection and result fusion during execution. The multi-model orchestration mechanism separates high-level planning (handled by general LLMs) from specialized execution (delegated to domain tools), ensuring both flexibility and analytical depth.

| Database/Tool | Function | Type |
|---|---|---|
| STRING | Protein interaction networks and functional associations | Database |
| BioGRID | Experimentally validated interactions | Database |
| UniProt | Protein sequences and annotations | Database |
| PDB | Three-dimensional structural data and protein complexes | Database |
| PubMed | Biomedical literature retrieval | Database |
| BLAST | Sequence similarity search across databases | Tool |
| PDBePISA | Interaction interface analysis | Tool |
| AlphaFold2-Multimer | Protein complex structure prediction | Tool |
| DockQ | Quality assessment of protein complex models | Tool |

Table 1: Biological Databases and Tools Integrated in the Intelligent Agent

### 2.2.3 RETRIEVAL-AUGMENTED KNOWLEDGE EXTRACTION

To address the challenge of extracting disease-associated genetic variants from distributed biomedical literature, we implement a Retrieval-Augmented Generation (RAG) system that enhances factual accuracy and temporal currency. The system follows a retrieve-then-generate paradigm: first performing real-time similarity searches across domain-specific knowledge bases, then conditioning

LLM generation on retrieved evidence. This approach mitigates hallucination risks while ensuring traceability to authoritative sources.

The RAG workflow involves three technical components: (1) *Knowledge preprocessing* chunks biomedical texts from PubMed and ClinVar into logical segments, encodes them using Qwen3 embeddings, and constructs a vector database for efficient similarity search; (2) *Query processing* combines semantic vector matching with keyword-based retrieval, followed by relevance reranking to prioritize functionally significant variants; (3) *Augmented generation* employs carefully designed prompt templates to guide a biomedical-tuned LLM to produce standardized variant annotations grounded in retrieved evidence, with explicit source attribution for verifiability. This system enables precise extraction of coding-region variants associated with specific diseases while maintaining audit trails to original literature.

### 2.3 BENCHMARK AND EXPERIMENT IMPLEMENTATION

#### 2.3.1 AGENT IMPLEMENTATION

PIPA is implemented in Python 3.11 with a modular architecture allowing component replacement and extension. The agent core utilizes deepseek-v3-671B with langgraph ai agent architecture. Tool interfaces are wrapped as RESTful APIs with standardized JSON schemas. The entire system can be deployed in containerized environments for scalable execution. All code and documentation will be made publicly available upon publication.

#### 2.3.2 EXPERIMENTAL VALIDATION PROTOCOLS

Co-immunoprecipitation assays were performed in HEK293 cells using FLAG-tagged DDRGK1 and GFP-tagged DNM2 constructs. Binding quantification utilized densitometric analysis of immunoblots with three biological replicates. Super-resolution microscopy employed multi-color SIM on COS7 cells with MitoDeepRed mitochondrial staining. Colocalization analysis used Pearson correlation coefficients from line profile measurements.

#### 2.3.3 BENCHMARK IMPLEMENTATION

The ER–MITO benchmark incorporates 110 mitochondrial outer membrane proteins with curated ER interactors. Mutation-disease associations were extracted through RAG-enhanced literature mining and manual curation. Evaluation metrics include coverage (proportion of correct associations retrieved), precision (accuracy of retrieved associations), and F1-score (harmonic mean of coverage and precision).

## 3 RESULTS

### 3.1 CASE STUDY: DNM2 PPI DISCOVERY AND INTERFACE PERTURBATION

To evaluate PIPA's capacity for mutation-aware PPI discovery and to uncover underexplored mutational effects, we applied the system to DNM2 as a representative case. Appendix 1-2 show the prompt and result of biology research interacting with PIPA. A quantitative criterion has been established for PIPA to evaluate and identify significant PPI changes—a metric that, to our knowledge, has not been proposed previously (in Appendix 3). **Interface Conservation Index (ICI).** The Interface Conservation Index (ICI) is defined as:

$$\text{Interface Conservation Index (ICI)} = \frac{\text{Residues}_{\text{mut}}^{\text{total}} + \text{H-bonds}_{\text{mut}}^{\text{total}} + \text{Saltbridges}_{\text{mut}}^{\text{total}}}{\text{Residues}_{\text{wt}}^{\text{total}} + \text{H-bonds}_{\text{wt}}^{\text{total}} + \text{Saltbridges}_{\text{wt}}^{\text{total}}} \tag{1}$$

The ICI formulation integrates multiple interface features through an average-weighted combination. Here, *Residues*, *H-bonds*, and *Saltbridges* denote the numbers of interface residues, hydrogen bonds, and salt bridges, respectively. Higher ICI values indicate greater preservation of the interaction interface after mutation.

To validate PIPA's predictive capabilities in a biologically relevant context, we applied the framework to DNM2 using the ICI threshold (in Appendix 4). Notably, although numerous pathogenic DNM2 mutations have been reported, only a small subset reside at protein–protein interfaces, highlighting an underexplored space with high potential for modulating DNM2's interactions and cellular functions. This integrative computational workflow allowed PIPA to autonomously prioritize interface mutations most likely to perturb ER-associated PPIs, providing mechanistic insights and identifying DNM2 as a protein with substantial untapped potential for further functional and therapeutic investigation, guiding the selection of targets for experimental validation. Co-immunoprecipitation (co-IP) assays in HEK293 cells confirmed the predicted interface perturbation: FLAG-DDRGK efficiently co-precipitated wild-type GFP-DNM2, whereas the E368K mutation substantially attenuated binding, validating disruption of the DNM2–DDRGK interaction. HEK293 cells lacking one component served as negative controls, confirming assay specificity 3.

### 3.2 ER–MITO BENCHMARK

To extend beyond the analysis of a single protein, we generalized the DNM2 workflow into a systematic benchmark suite focused on PPI identification and disease-related mutation retrieval for mitochondria–ER. This suite is designed to evaluate large language models and agents across complex tasks, reflecting different levels of biological reasoning and researching ability.

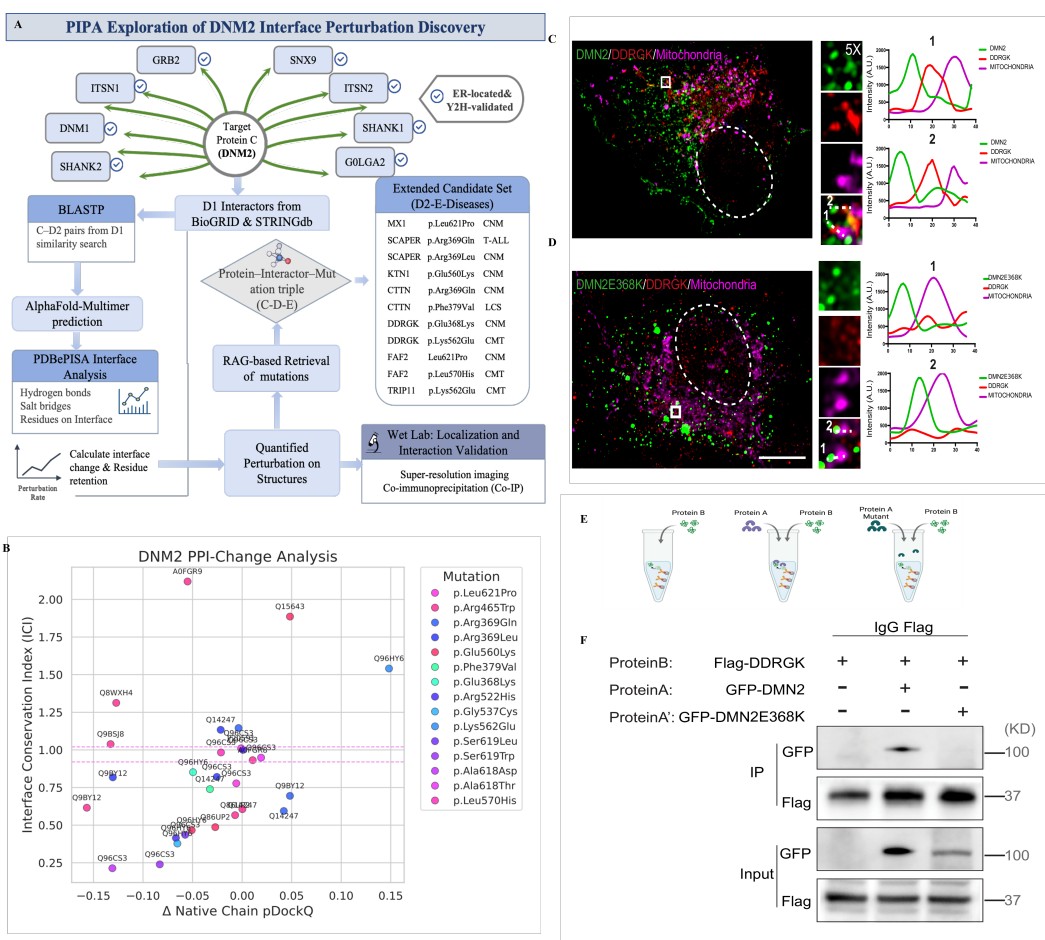

Figure 3: Exploration and experimental validation of DNM2–DDRGK1 interface perturbation. (A) the interactome identified for DNM2, (B) the metric plot for mutants.(C,D) Super-resolution microscopy reveals altered subcellular localization caused by alterations in PPIs.(E,F) Co-immunoprecipitation shows reduced binding for E368K mutant.

**Task 1: Prediction of Novel Interacting Proteins.** To evaluate AI systems in the context of mitochondria–ER protein networks, we established a systematic benchmark suite encompassing two complementary tasks. The first task focuses on the discovery of potential inter-organelle protein interactions. Starting from a curated set of 110 mitochondrial outer membrane proteins, candidate ER-localized interactors lacking prior yeast two-hybrid validation were identified. Models are tasked with predicting both the existence of interactions and the residues mediating them at the interface, emphasizing biologically plausible PPIs that bridge mitochondria and ER and providing mechanistic insight into inter-organelle functional crosstalk.

**Task2 : Protein–Disease Mutation Retrieval.** The second task extends the framework to systematically explore disease-associated variation, requiring models to link individual proteins to pathogenic mutations. ER–MITO proteins were categorized into three groups: (i) mitochondrial outer membrane proteins, (ii) ER-localized proteins, and (iii) proteins co-localized at mitochondria–ER contact sites. Using a retrieval-augmented generation (RAG) strategy complemented by manual curation, pathogenic mutations were extracted from databases and literature, standardized according to HGVS protein nomenclature, and mapped onto their corresponding proteins. This benchmark evaluates the model's ability to integrate mutation data with interface-aware predictions, providing a mechanistic connection between pathogenic variants and ER–mitochondria interactions, and highlighting mutations with potential functional impact.

The benchmark results reveal significant performance variations across different models. In Task 13, GPT-5.0 achieves the highest protein-level F1 score (11.92%) but demonstrates poor residue-level performance (0.26% F1), indicating limitations in fine-grained interaction prediction. Qwen3-235B shows balanced performance with moderate scores at both levels.

For Task 2**??**, Biomni achieves the best overall performance (3.52% F1), suggesting stronger capabilities in mutation retrieval tasks. The generally low scores across all models highlight the challenge of accurate PPI prediction and mutation annotation, emphasizing the need for continued methodological improvements in biological AI systems.

Notably, all models perform better at protein-level prediction compared to residue-level analysis, reflecting the increased complexity of interface residue identification. The benchmark establishes a foundation for systematic evaluation of AI capabilities in computational biology.

Table 2: Benchmark1 results of different models (%)

| Model | Subset | Avg Coverage | Avg Precision | Avg F1 |
|---|---|---|---|---|
| Biomni | Protein | 3.41 | 9.52 | 5.02 |
| | Residue | 1.65 | 7.74 | 2.37 |
| GPT-5.0 | Protein | 19.72 | 8.54 | 11.92 |
| | Residue | 0.22 | 0.57 | 0.26 |
| DeepSeek V3 | Protein | 3.66 | 7.75 | 4.97 |
| | Residue | 0.03 | 0.31 | 0.04 |
| DeepSeek R1 | Protein | 2.77 | 15.68 | 4.70 |
| | Residue | 0.00 | 0.27 | 0.00 |
| Qwen3-235B | Protein | 6.09 | 13.40 | 8.38 |
| | Residue | 0.07 | 0.94 | 0.12 |
| Qwen2.5-72B | Protein | 0.71 | 2.28 | 1.08 |
| | Residue | 0.04 | 0.56 | 0.07 |

Table 3: Benchmark2 results of different models (%)

| Model | Avg Coverage | Avg Precision | Avg F1 |
|---|---|---|---|
| Biomni | 3.77 | 4.45 | 3.52 |
| GPT-5.0 | 0.58 | 1.74 | 0.73 |
| DeepSeek V3 | 0.35 | 0.74 | 0.40 |
| DeepSeek R1 | 2.65 | 3.01 | 2.19 |
| Qwen3-235B | 1.46 | 1.24 | 1.13 |
| Qwen2.5-72B | 0.02 | 0.29 | 0.04 |

## 4 RELATED WORK

### 4.1 PPI IDENTIFICATION USING ALPHAFOLD2

The breakthrough of AlphaFold2 (AF2) (Jumper et al., 2021b) revolutionized structural biology by providing high-accuracy protein structure predictions, which naturally extended to protein-protein interaction (PPI) studies. Subsequent work demonstrated that AF2's architecture could be adapted for direct PPI prediction. Gao et al. (2022b) introduced AF2Complex, which leverages the core AF2 network to predict physical interactions in multimeric proteins without requiring paired multiple sequence alignments, significantly outperforming both AF-Multimer and traditional docking methods. To address the need for reliable confidence estimation, Bryant et al. (2022b) developed DockQ and pDockQ metrics, later refined as pDockQv2 (Zhu et al., 2023), providing robust assessment frameworks for predicted complexes. These developments established the structural paradigm for PPI identification, enabling accurate modeling of protein interfaces at scale. However, AF2 exhibits limitations in predicting *de novo* interactions lacking evolutionary traces, such as antibody-antigen binding (Smith et al., 2025), prompting innovations that integrate experimental data. For instance, Drake et al. (2025) proposed DMS-Fold, which incorporates residue burial restraints from deep mutational scanning to refine AF2 predictions, improving accuracy for 88

### 4.2 MACHINE LEARNING WITH ALPHAFOLD2-DERIVED FEATURES

Parallel to direct structural prediction, classical machine learning approaches have exploited AF2-generated structures to engineer informative features for PPI characterization. Early efforts included ppdx (Conti et al., 2022), which automated the computation of 95 interaction descriptors, and HCML (Zhou et al., 2022), which combined persistent homology with gradient boosting to predict mutation-induced affinity changes. Whole-proteome scale prediction was advanced by PEPPI (Bell et al., 2022a), integrating structural similarity, functional associations, and naive Bayes classification. More recently, Mischley et al. (2024) and Schmid and Walter (2025) combined AF2-derived structural features with supervised classifiers to distinguish true interactions from false positives in large-scale screens, highlighting the relevance of feature-based methods for interpretable PPI inference. These approaches excel in leveraging structural constraints but often rely on handcrafted feature engineering, limiting their adaptability to complex interface dynamics.

### 4.3 DEEP LEARNING ARCHITECTURES FOR PPI PREDICTION

Deep learning models, particularly graph neural networks (GNNs), have enabled end-to-end PPI prediction by learning directly from structural or sequence data. Representative works include EGRET (Mahbub and Bayzid, 2022), which employed edge-aggregated graph attention networks with transfer learning; Struct2Graph (Baranwal et al., 2022) and DeepRank-GNN (**?**), which optimized GNNs for interface residue prediction; and DockNet (Williams et al., 2023), a siamese GNN for contact prediction. Sequence-based models like SDNN-PPI (Li et al., 2022) and ADH-PPI (Asim et al., 2022) explored hybrid architectures combining CNNs, LSTMs, and self-attention. Contemporary methods address specific challenges: Gated-GPS (Gao et al., 2025) tackles class imbalance via Graph Transformers and gating mechanisms, while CollaPPI (Ma et al., 2024) introduces collaborative learning across protein and task hierarchies. These approaches excel in leveraging structural constraints but often rely on handcrafted feature engineering, limiting their adaptability to complex interface dynamics.

### 4.4 AGENTIC SCIENTIFIC DISCOVERY

The latest frontier involves AI agents that automate multi-step scientific discovery workflows. Bind-Craft (Correia et al., 2025) exemplifies this shift, enabling *de novo* design of functional protein binders via AF2 weight optimization and experimental validation (46.3 % success rate). It autonomously handles target flexibility, loss function optimization, and *in silico* filtering, demonstrating "one-shot design" for targets like PD-1/PD-L1 and CRISPR-Cas9. Beyond protein design, agentic systems automate experimental workflows; for instance, language-model-driven agents have been deployed at particle accelerators to execute multi-stage experiments, reducing preparation time by two orders of magnitude (Brown et al., 2025). The Biomni platform (Lee et al., 2025) further generalizes this paradigm, integrating a unified biomedical environment with agent architecture

to autonomously perform tasks like variant prioritization and drug repositioning. GPT-5's reasoning advances (OpenAI, 2025)—such as dynamic routing and reduced hallucination rates—provide foundational support for complex task orchestration. These developments signal a transition toward AI-driven discovery ecosystems, where agents *orchestrate* integrate predictive and generative tools into coherent pipelines, minimizing human intervention.

## 5 CONCLUSION AND FUTURE WORK

We presented PIPA, an intelligent agent that automates the discovery and characterization of protein-protein interactions and their pathological perturbations. PIPA integrates structural bioinformatics, database mining, and mutational analysis through a unified tool-augmented architecture, demonstrating three key innovations: (1) PIPA provides a scalable framework for autonomous scientific investigation, with the Plan-Select-Execute-Reflect loop that ensures robust and interpretable workflow execution, while its hierarchical tool abstraction effectively manages the complexity of heterogeneous biological data and tools; (2) it introduces a hierarchical tool abstraction layer that seamlessly integrates heterogeneous resources—including biological databases, computational tools, and RAG system for literature mining—into a unified interface, enabling coherent multi-modal data integration and (3) the ER–MITO benchmark for rigorous evaluation of PPI identification and pathogenic mutation retrieval addresses a critical community need by enabling the quantitative evaluation of scientific reasoning in AI systems. PIPA's effectiveness was validated through both computational experiments and wet-lab validation of the DNM2–DDRGK1 interaction. PIPA represents a step toward autonomous biological discovery systems, laying the groundwork for future extensions to other biomedical domains.

## 6 ETHICS STATEMENT

This work presents a computational framework, PIPA, for predicting and analyzing protein-protein interactions (PPIs) and the effects of mutations. All data used in this study were sourced from publicly available databases (RCSB PDB, BioGRID, STRING, etc.) and scientific literature. The experimental validation involving cell cultures (HEK293, COS7) was conducted following standard ethical guidelines and biosafety protocols. We do not foresee any immediate direct misuse of our technology. Our work is intended solely for scientific research, such as understanding disease mechanisms and accelerating therapeutic discovery. The authors declare no competing interests.

## 7 REPRODUCIBILITY STATEMENT

To ensure the reproducibility of our work, we have made substantial efforts to document our methodology and provide access to key resources. The complete source code for the PIPA agent framework, including the tool orchestration logic, data processing pipelines, and the implementation of the PPI identification and perturbation, will be released under an open-source license upon publication. All software dependencies and version numbers are listed in the appendix. The commands and parameters for running AlphaFold-Multimer predictions and PDBePISA analyses are explicitly stated in the Methods. The scripts used to generate the results and figures from the processed data are also included in the supplementary code repository. We believe these measures provide the necessary details for replicating the core findings of this study.

## 8 LLM USAGE

This work utilized GPT-4 as a writing assistant for language polishing and editing. The model helped refine technical descriptions and improve narrative flow.

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
