# OpenReview forum: "PIPA: An Agent for Protein Interaction Identification and Perturbation Analysis"
_ICLR.cc/2026/Conference — ICLR 2026 Conference Desk Rejected Submission_

### Official Review · Reviewer_8CLf · 2025-10-26

**Soundness:** 2
**Presentation:** 4
**Contribution:** 2
**Rating:** 4
**Confidence:** 2

**Summary:**

The paper presents a tool called PIPA, which automates the discovery and analysis of protein-protein interactions and their pathological perturbation. It provides a scalable framework by integrating heterogeneous resources such as biological databases, AlphaFold-based structure prediction tools. The system is evaluated on predicting the inter-organelle protein interaction task and the protein to pathogenic mutation prediction problem.

**Strengths:**

- The paper is clearly written, with illustrative figures showing the workflow and architecture of the proposed tool.
- The paper provides a unified framework that combines diverse data sources such as protein databases, literature mining, and leverages large language models for scientific reasoning.

**Weaknesses:**

- The main contribution of the paper lies in system integration rather than in developing new learning methods or representations. While the system design is ambitious, most components are existing tools or standard techniques.
- The benchmark results show relatively low scores across models.
- Practical aspects such as computational resources, runtime efficiency, and hardware requirements are not discussed.

**Questions:**

- The paper mentions using Qwen3 embeddings for constructing the vector database in the RAG pipeline. Could the authors elaborate on why this specific model was chosen over other possible alternatives? Since embedding choice can affect retrieval quality, a brief comparison could help assess robustness.
- The description of the knowledge preprocessing step does not specify the chunk size or overlap used when segmenting biomedical text. Could the authors clarify these parameters and how they were optimized for biomedical contexts?
- What are the hardware and software requirements to run PIPA end-to-end?
- The reported F1 scores across all models (Table 2–3) are low. Do the authors attribute this mainly to (a) model limitations (i.e., insufficient biological reasoning in current LLMs), (b) the benchmark's difficulty or design, or (c) limitations in the orchestration pipeline?

**Additional comments and typos**
- In the appendix, it seems that prompts must start with “You are a bioinformatics assistant,” which might be cumbersome for users. A higher-level interface or predefined workflow templates might simplify user interaction.
-  In Line 342, Task 13 -> Task 1 (i.e. Table 2)
- in Line 345, “For Task2??” -> “For Task2”
- In Line 413, missing reference

---

### Official Review · Reviewer_kJ7x · 2025-10-31

**Soundness:** 1
**Presentation:** 1
**Contribution:** 2
**Rating:** 0
**Confidence:** 4

**Summary:**

The authors present an intelligent agent, "PIPA", which is a DeepSeek-based model complemented with the LangGraph framework for stateful agents. Their model orchestrates the combination of existing bioinformatics tools (AlphaFold-multimer, BLAST, DockQ) and databases (BioGRID, Uniprot, PDB, etc.) in a sequential fashion.

The authors pair their model with a RAG pipeline to augment its knowledge base with information contained in PubMed and ClinVar. Their hope is that their model can assess the structural impact of mutations and their consequence on protein-protein interactions and disease.

The dynamin-2 protein is used to demonstrate how their model can predict the disruption of an interaction following a mutation and present some in vitro data to validate their prediction.

The authors then evaluate their model for the tasks of predicting 1) new PPIs/residues at the interface for inter-organelle proteins and 2) the impact of mutations on PPIs for proteins located on the surface of the endoplasmic reticulum and mitochondria.

**Strengths:**

- The idea of orchestrating several bioinformatics tools using an AI agent is interesting. There definitely is a use for such a tool.

- The overall idea is interesting.

**Weaknesses:**

**Methodology**:
- In vitro experimentation: Insufficient information to replicate findings. Need to include *full* methods in supplementary information.
- In the "Reproducibility statement", some statements are incorrect, including: "All software dependencies and version numbers are listed in the appendix. The commands and parameters for running AlphaFold-Multimer predictions and PDBePISA analyses are explicitly stated in the Methods." I could not locate that information.
- It is not clear what the models are predicting in Task 2 (mapping a mutation to a loss of PPI, a disease?). It is also not clear how large the test sets for "Task 1" and "Task 2" are and how they were assembled/curated. Where are those 110 (mito-ER) PPIs extracted from? Are they structures (needed for interacting residues ground truth)? How did you ensure the data was accurate ("manual curation") for disease-causing mutations? How many did you end up with? The accuracy metrics provided are difficult to trust and contextualized with such vague descriptions of the test sets...


The paper's overall **presentation** requires heavy revisions:
- Overall impression: the paper was difficult to follow and logically poorly organized.
- It is unclear why "Related works" is kept for the end of the paper when it provides contextual information that is useful for the interpretation of the paper
- Inconsistent use of the words "benchmarks" and "tasks" is confusing.
- Odd formatting of cross-references (e.g., "3", instead of "Table 3"). Some cross-references and references are not resolved by LaTeX and are replaced with bold question marks (?)
- Figures/flow charts are difficult to interpret... I suspect they were largely AI-generated. Icons used within the diagrams are often nonsensical.
- Section 4.1 ends with a truncated sentence.
- Last sentence of section 4.3 is copied and pasted from section 4.2...
- Capitalization of reference titles is incorrect (in the References section)
- Figure 2 is compressed... (incorrect aspect ratio)
- Why is the "I" in "PIPA" in italics in the title, but not elsewhere?
- Qwen3 embeddings (arXiv:2506.05176v3) should be properly cited. The pre-print's latest revision was uploaded in 2025 and is not in widely known/adopted.
- The first sentence of the Introduction refers to ideas introduced in the abstract, which should be standalone. ("interaction interface" used without context)

**Questions:**

- In the formulation of the ICI, the number of residues appears in the equation, but in contrast with the number of H-bonds and salt bridges, it is not clear what that means... The number of residues does not change with missense mutations... Is "Residues^total" the number of residues at the interface? If so, how is that defined (e.g., residues within a distance "d" from the interaction partner)?

- Why is the threshold on ICI in the main text (line 271), but on pDockQ in Appendix 4?

- Where is PIPA in the benchmark analysis???

**Details Of Ethics Concerns:**

Use of AI should be disclosed with transparency. The authors state that they only used AI to improve the narrative flow of their paper, but I have reasons to believe that generative AI was used to generate Figures 1 and 2, partially or in their entirety. I suspect this for the following reasons:

- Non-sensical/meaningless icons (e.g., Figure 1: next to tool/database names, Figure 2: the blobs under "Predicted interactions", the odd protein cartoon under "Predictzed protein interactions")
- Obvious typos such as "Intential", "divergenzzcy", or "predictzed" (Figure 2)
- Odd reverse commas (Figure 2)
- Odd alignment of icons with text under "Evaluation metrics" (Figure 2)

The figures are poorly designed and ineffective, regardless of whether AI was used to generate them or not.

---

### Official Review · Reviewer_ZdPd · 2025-10-31

**Soundness:** 2
**Presentation:** 2
**Contribution:** 1
**Rating:** 2
**Confidence:** 4

**Summary:**

The authors develop an agentic workflow for identifying mutations that disrupt protein-protein interaction interfaces. It is unclear what is the purpose of the agentic workflow, since all of the important bioinformatic steps of the pipeline seem to be clearly defined by the authors and fed directly into the AI prompts. It is unclear what is being automated here at all that could not be done by a series of scripts or a Snakemake workflow. Despite this, the AI is obviously unable to complete the task put to it; the appendices report 27 failures and hallucinations that required human correction compared with just 5 successes. There is no evidence in the paper that the agentic workflow is (on its own, without manual curation) able to identify or prioritize pathogenic mutations.

- The authors develop an "Interface Conservation Index", but it is never made clear how interface residues are selected. In addition, this new metric should be compared to something more standard like an interface RMSD or fraction of native contacts from the wild type
- There is no information about how the ER-MITO benchmark was curated, nor any descriptive statistics about this benchmark.
    - Task 1 and 2 of this benchmark are not clearly defined, nor is it clear how evaluations are performed. Biomni, itself an agentic workflow, is compared with LLMs, presumably used within the authors workflow-- but Biomni appears to be the best performing.
    - Across all models, the absolute performance is incredibly low
- The experimental protocol is incredibly sparse and not nearly descriptive enough to allow anyone to attempt to replicate these experiments.
    - Also, why was the E368K mutation selected? By the ICI plot in Fig 3B, it doesn't seem to clearly stand out from the others.
- Figures contain obvious errors or inconsistencies and lack descriptive captions. This makes it challenging to follow even what the author's proposed method is.
    - "Uniport" in Figure 1, what is "LAST"?
    - The protein chain in Figure 1 has random tangles of loop?
    - "potential intential interacting, divergenzzcy, predictzed interface" in Figure 2
    - In Figure 3A, why does calculating interface change occur prior to RAG-based retrieval of mutations? Without proposed mutations how can one quantify the effect of that mutation? There is also a random line in Fig3A that connects nothing
    - Fig 3C and D are missing X units labels, descriptions of the subfigures, indication of what (1) and (2) are, or anything else that aides in properly interpreting these results. Also, the key protein is referred to as DMN2 in some places and DNM2 in others
- Numbers appear randomly throughout the text, e.g. L068 "...interface analysis 2", L280 "...confirming assay specificity 3", L394 "...improving accuracy for 88"
- BindCraft is not an agentic workflow, as indicated in Section 4.4
- References are completely messed up:
    - Author names are inlined without parentheses for citations, for example "...via AlphaFold-Multimer Evans et al. (2022)"
    - Some citations are simply wrong-- Correia is the last author of BindCraft, and Grace Lee is not an author of Biomni
    - Bryant et al., Jumper et al., and Bell et al. all appear twice in the citations
    - The senior author of Jha et al is Sourav Karmakar, not Sudipta
    - Jane Smith et al does not exist,although this appears to be referencing doi: 10.1016/j.tibtech.2025.04.013
    - I'm sure there are other issues that I'm not familiar enough with the original paper to catch
    - Some LaTeX references are completely missing, as ??

**Strengths:**

see above

**Weaknesses:**

see above

**Questions:**

see above

---

### Official Review · Reviewer_fuqF · 2025-11-01

**Soundness:** 2
**Presentation:** 2
**Contribution:** 1
**Rating:** 2
**Confidence:** 2

**Summary:**

This paper introduces PIPA, a tool-augmented agent designed to automate the end-to-end scientific discovery pipeline for protein-protein interactions (PPIs) and their pathological perturbations. PIPA integrates a large language model core for reasoning within a "Plan-Select-Execute-Reflect" loop, orchestrating a diverse suite of specialized bioinformatics tools (e.g., AlphaFold-Multimer, BLAST, PDBEPISA) and databases (e.g., STRING, PubMed via RAG). The authors demonstrate PIPA's capability through a compelling case study on the DNM2 protein, where the agent autonomously hypothesized an interaction and a specific perturbation that was subsequently validated in a wet lab. The paper also contributes ER-MITO, a new benchmark dataset for evaluating an agent's ability to predict inter-organelle PPIs and retrieve disease-associated mutations.

**Strengths:**

Here are some of the strengths of the paper:

1. $\textbf{Real-World Validation}$

The agent presented in the paper has already been proven useful in a real-world lab, which is a compelling reason for its adoption.

2.  $\textbf{Benchmark Proposal}$

The contribution of the ER-MITO benchmark could be significant. It addresses a critical need for rigorous, domain-specific evaluation of AI agents in science. The benchmark's two-task structure (PPI prediction and mutation-disease retrieval) tests complex reasoning and tool use, moving well beyond simple question-answering.

3. $\textbf{Tool Orchestration}$

The paper presents a strong engineering achievement in integrating a heterogeneous set of specialized tools. Successfully managing the inputs and outputs of database APIs, sequence aligners (BLAST), structure predictors (AlphaFold-Multimer), interface analyzers (PDBEPISA), and others within a single agentic framework is a non-trivial and necessary step for building autonomous scientific agents.

**Weaknesses:**

Here are some weakensses of the paper:

1. $\textbf{Limited ML Novelty}$

Even if the presented work is important, it seems it fits more as a software engineering accomplishment rather than an ML accomplishment, which limits the novelty of the proposed work.

2. $\textbf{Limited Expemental Evaluation}$

Are the results in Table 2 and Table 3 evaluations of PIPA? Or are they direct LLM evals? The results also seem very low. Is that normal? Is there a way to compare to other methods in addition to presented results?

**Questions:**

Please look at the weaknesses section

---

### Note · Program_Chairs · 2026-01-17
**Submission Desk Rejected by Program Chairs**

The following references in this submission do not refer to real documents and/or have major errors in bibliographic information:

 Alex Brown et al. Autonomous scientific experimentation at the advanced light source using language-model-driven agents. Nature Communications, 16:7001, 2025.